# Current Evidence on the Role of the Gut Microbiome in ADHD Pathophysiology and Therapeutic Implications

**DOI:** 10.3390/nu13010249

**Published:** 2021-01-16

**Authors:** Ana Checa-Ros, Antonio Jeréz-Calero, Antonio Molina-Carballo, Cristina Campoy, Antonio Muñoz-Hoyos

**Affiliations:** 1School of Life and Health Sciences, Aston University, Birmingham B4 7ET, UK; 2Department of Clinical Neurophysiology, Birmingham Women’s and Children’s NHS Foundation Trust, Birmingham B4 6NH, UK; 3Department of Pediatrics, School of Medicine, University of Granada, 18016 Granada, Spain; amolinac@ugr.es (A.M.-C.); ccampoy@ugr.es (C.C.); amunozh@ugr.es (A.M.-H.); 4Department of Pediatrics, San Cecilio University Hospital, 18016 Granada, Spain; aejerezc@gmail.com

**Keywords:** gastrointestinal microbiome, ADHD, circadian rhythm, fatty acids, omega-3, probiotics

## Abstract

Studies suggest that the bidirectional relationship existent between the gut microbiome (GM) and the central nervous system (CNS), or so-called the microbiome–gut–brain axis (MGBA), is involved in diverse neuropsychiatric diseases in children and adults. In pediatric age, most studies have focused on patients with autism. However, evidence of the role played by the MGBA in attention deficit/hyperactivity disorder (ADHD), the most common neurodevelopmental disorder in childhood, is still scanty and heterogeneous. This review aims to provide the current evidence on the functioning of the MGBA in pediatric patients with ADHD and the specific role of omega-3 polyunsaturated fatty acids (ω-3 PUFAs) in this interaction, as well as the potential of the GM as a therapeutic target for ADHD. We will explore: (1) the diverse communication pathways between the GM and the CNS; (2) changes in the GM composition in children and adolescents with ADHD and association with ADHD pathophysiology; (3) influence of the GM on the ω-3 PUFA imbalance characteristically found in ADHD; (4) interaction between the GM and circadian rhythm regulation, as sleep disorders are frequently comorbid with ADHD; (5) finally, we will evaluate the most recent studies on the use of probiotics in pediatric patients with ADHD.

## 1. Introduction

Billions of microorganisms inhabit the human body (“microbiota”), including bacteria, archaea, fungi, viruses and protozoa. They and their genes (“microbiome”) are involved in different biological functions, some of them are essential for our survival [1,2,3]. in particular, the microbiota living in the digestive tract is composed of more than 10^4^ microorganisms from 300–3000 different species, which encode 200 times the number of human genes [4,5]. The gut bacteria mainly include six major phyla of Firmicutes, Bacteroidetes, Proteobacteria, Actinomycetes, Verrucomicrobia and Fusobacteria, with Bacteroidetes and Firmicutes as the dominant ones [6]. This rich and diverse gut microbiome is of utmost importance, as it has far-reaching implications in a variety of gastrointestinal and non-gastrointestinal functions. They participate in the metabolism and absorption of nutrients, including carbohydrates and proteins, bile acid, vitamins and other bioactive compounds [3,7]. Regarding the non-gastrointestinal functions, the gut microbiome has been reported to impact on the brain development [8,9,10,11] and the maturation of the immune [12] and neuroendocrine systems [13,14]. As a matter of fact, this impact takes place only during a critical period in growth and development and it cannot be reversed afterwards [12,15]. The effects caused by the gut microbiome begin even before birth, as the embryonic development is also influenced by the maternal gut microbiome [16].

The relationship between the gut microbiome and their host is bidirectional [17], so that human beings also regulate the composition and number of the microorganisms that colonize the digestive tract through factors related to diet, health and lifestyle. Modernization and its consequent changes in the human diet, including dietary patterns, habits and food processing, have greatly influenced the gut microbiome [18,19,20]. Modernization has also changed the delivery mode, with an increasing number of women undergoing caesarian sections, which has had an impact on the composition of the commensal microbiome [21]. Another major influence is exerted by modern advances in medicine and the ongoing development of new pharmacological treatments, particularly those for chronic conditions [22].

Modern life and the consequent impact on the gut microbiome have also made fundamental changes in the pattern of human illnesses, which has shifted from traditional infectious diseases towards increasingly frequent autoimmune diseases, such as asthma and allergies; cardiovascular diseases, such as hypertension; metabolic diseases, like diabetes; mental diseases, such as depression and anxiety; and a variety of neurological and neurodevelopmental disorders, such as multiple sclerosis, Alzheimer’s, Parkinson, autism spectrum disorders (ASD) and attention-deficit and/or hyperactivity disorder (ADHD). All of these conditions have been associated with an imbalance in the gut microbiome composition [23,24].

In this review, we will focus on the participation of the gut microbiome into the pathophysiological mechanisms of ADHD and the therapeutic potential of these microorganisms in pediatric patients with this disorder.

## 2. Gut Microbiome and Neurodevelopment: The Gut–Brain Axis

The term “gut–brain axis” has been coined to describe the bidirectional communication between the gut microbiome and the central nervous system (CNS) [25,26]. This axis is also referred to as the “microbiota–gut–brain axis” (MGBA), to emphasize the participation of the gut microbiota in this interaction [27]. Three main pathways configure this axis: the nerve pathway, the neuroendocrine pathway and the immune pathway (Figure 1).

### 2.1. Nerve Pathway

The gut is innervated by the hepatic and celiac branches of the vagus nerve. Depending on their location and type, vagal afferents detect a variety of mechanic (stretch, tension) and chemical stimuli (bacterial by-products, gut hormones, neurotransmitters) [28]. The vagus nerve plays a substantial role in mood regulation. Examples of this are the therapeutic use of vagus nerve stimulation in refractory depression [29] and chronic pain [30], probably associated with a modulation of catecholamine release in brain regions related to anxiety and depression [29]. A recent study in murine models reported that activation of gastrointestinal vagal afferents influence reward behavior [31]. The gut microbiome has the capacity to modulate the host’s emotional and behavioral responses by acting on the vagal afferents. In animal models, infections by pathogens like *Campylobacter jejuni* and *Citrobacter amalonaticus* induced anxiety-like behavior [32], whereas supplementation with probiotics including *Lactobacillus rhamnosus* and *Bifidobacterium longum* alleviated these anxiety/depression-like conducts [33,34]. Interestingly, the behavioral effects induced by *Lactobacillus reuteri* in genetic mouse models of autism were halted in vagotomized mice [35]. 

Another important network of neurons lying between the microbiota and the host is configured by the enteric nervous system (ENS), which is composed of the myenteric and submucosal plexus. The ENS communicates with the CNS via afferent neurons with sensory information that follow spinal and vagal routes, and it is responsible for the coordination of gut functions, such as motility and fluid movement [36]. Maturation and functions of the ENS seem to be influenced by the gut microbiome. Germ-free (GF) mice displayed significant abnormalities in the ENS structure and neurochemistry in the early postnatal period, which disappeared after colonization [37]. The pathways through which the gut microbiome plays a role in the ENS are yet to be clarified. They may involve the activation of pattern recognition receptors (PRRs), including Toll-like receptors (TLRs) [38], and the expression of serotonin (5-HT_4_) receptors [39].

The gut microbiome may impact on the generation of neurotransmitters, either by synthesizing them “de novo” or by influencing the neurotransmitter-related metabolism pathways. Bacterial species of *Clostridium perfringens* modulates the synthesis of 5-HT via the expression of tryptophan hydroxylase-1, which is the rate-limiting enzyme in its synthesis [40]. In fact, concentrations of this neurotransmitter, which is involved in socio-affective processing, anxiety and fear [41,42], were found significantly reduced in GF mice [43]. As 5-HT impacts on gut motility, it can now be suggested that the gut microbiome regulates intestinal motility through the serotonergic system [44]. The gut microbiome also influences dopamine (DA) levels in the frontal cortex and striatum in rodents, two brain areas involved in executive functions [45]. It has been extensively reported that disruptions in dopaminergic and serotonergic systems are associated with the appearance of mood-related disorders and cognitive functions [46,47]. Therefore, the connections established between these neurotransmitters and the gut microbiome may provide evidence for the role of these microorganisms in the pathophysiology of diverse neuropsychiatric disorders, such as anxiety, depression, bipolar disorder and neurodevelopmental disorders, such as ASD and ADHD [48,49]. Other neurotransmitters, such as glutamate (Glu) and gamma-aminobutyric acid (GABA), are also synthesized by the gut microbiota [50,51]. In a recent systematic review, gut microbiota including *Campylobacter jejuni* and *Bacteroides vulgatus* was found to influence cognitive function in patients with Alzheimer’s disease via Glu metabolism [52].

The intestinal microbiome is also involved in the kynurenine pathway, which is in turn implicated in neuroinflammation processes associated with schizophrenia and depression [53]. Mice that developed symptoms of schizophrenia after being transplanted with fecal microbiota from drug-free patients with schizophrenia, were found to exhibit increased levels of kynurenic acid in periphery and brain [54]. A reduced conversion of tryptophan into kynurenine metabolites have been found in GF mice [55]. The interaction between the gut microbiome and the enzymes involved in the kynurenine pathway (indoleamine-2,3-dioxygenase and tryptophan-2,3-dioxygenase) seem to be mediated through the immune responses to stress [56] and redox imbalance [57]. Indeed, a correlation was found between reduced concentrations of *L. reuteri* and increased serum levels of kynurenines in mice exposed to chronic stress, which was secondary to bacterial regulation of kynurenine enzymes via the production of H2O2 [58].

As reported in animal studies, the gut microbiome regulates the expression of the brain-derived neurotrophic factor (BDNF), which is involved in neurogenesis [59,60]. In adults with mild cognitive impairment, cognitive and attentional enhancement were reported after the administration of *Lactobacillus plantarum* for 12 weeks, which were associated with an increase in BDNF levels [61]. Lower levels of myelination in total brain and major grey and white matter structures at either 4 or 12 weeks of age were also found in GF mice [62]. In a mouse model of multiple sclerosis, fecal microbiota transplantation rebuilt the gut microbiome and provided therapeutic benefits by conferring protection on the myelin, with additional effects on the blood–brain barrier and the populations of astrocytes [63]. The apoptosis and neurodegeneration seem to be influenced by the gut microbiome, as neonatal GF mice exhibited increased apoptosis in the hypothalamus and the hippocampus in comparison with conventional mice [64]. Alterations of the gut microbiome may additionally lead to the promotion of amyloid formation [65], whereas probiotic supplementation can prevent or even stop this process [18,66].

### 2.2. Neuroendocrine Pathway

The gut microbiome is essential in the development and function of the hypothalamic–pituitary–adrenal (HPA) axis, which represents the crux of the neuroendocrine transmission and the stress response system [14]. In GF mice, the HPA axis response was exaggerated and the sensitivity to negative feedback signals was reduced. The administration of *Bifidobacterium infantis* at an early stage reversed this response [13,14]. In human patients with irritable bowel syndrome, exaggerated adrenocorticotropic hormone (ACTH) and cortisol responses were observed after infusion of corticotropin-release factor (CRF) [67], together with a dysfunctional microbiota [68]. The gut microbiome directly influences the production of glucocorticoids and immune mediators, such as tumor necrosis factor-alpha (TNF-α) and interleukins-1beta and 6 (IL-1β and IL-6), which in turn stimulates the HPA axis [69].

The gut microbiome modulates the secretion of gastrointestinal peptides that mediate metabolic functions related to energy homeostasis, such as insulin, leptin, ghrelin, neuropeptide Y family (NPY) and glucagon-like-peptide 1 (GLP-1) [70]. Microbial disturbances induce insulin resistance, which is alleviated after microbial restoration [71,72]. The NPY family consists of different neuropeptides involved in energy homeostasis, mood and stress responses [73]. The gut microbiome recognizes NPYs and modulates their synthesis and secretion [73,74]. Oxytocin participates in a variety of activities, such as parturition, lactation, social interaction and stress response [75]. Offspring born to mothers taking a high-fat diet exhibited social deficits, abnormalities in the gut microbiota and a reduction in the hypothalamic neurons immunoreactive to oxytocin. These alterations could be prevented by cohousing with offspring of mothers on a regular diet or by treatment with *L. reuteri* [76]. Functions of the endogenous opioid system are essential in analgesia, tolerance and dependence [77]. Opioids have the capacity to alter gut microbial diversity, whereas the gut microbiome mediates tolerance to their analgesic effects [78,79].

The intestinal microbiome produces short-chain fatty acids (SCFAs), particularly butyrate, propionic acid and acetate, mostly derived from the degradation of fibres and undigested saccharides [80]. They serve as important mitochondrial fuels, particularly under conditions of inflammation, starvation and physical strain [81,82]. In animal studies using GF and specific-pathogen free (SPF) mice, SCFAs were reported to affect mitochondrial energy metabolism through a variety of transcription factors, such as the peroxisome proliferator-activated receptor gamma coactivator 1-alpha (PGC-1α) and the peroxisome proliferator-activated receptor gamma (PPAR-α) [83,84,85]. Although the mechanisms are yet to be unraveled, SCFAs also may combat the oxidative stress which is predominant during proinflammatory states by upregulating the activity of antioxidant enzymes, like glutathione peroxidase (GPx) and catalase [86]. Additionally, butyrate has anti-inflammatory properties, as it induces the release of the anti-inflammatory IL-10, whereas inhibits proinflammatory cytokines (IL-6, IL-12) [87]. In a mouse model of depression, reduced fecal concentrations of SCFAs were found. Administration of the carotenoid crocin-I for 6 weeks improved symptoms by alleviating the gut microbial dysbiosis and increasing SCFAs in feces [88].

The gut microbiome also creates other substances derived from the metabolism of proteins and amine acids, such as ammonia (NH3) [89], which in excessive levels (hyperammoniemia) represents an important risk factor for neurological diseases, like hepatic encephalopathy and autism [90].

### 2.3. Immune Pathway

The development and integrity of the gut barrier and the blood-brain barrier (BBB) are dependent on the gut microbiome. Alterations of the gut microbiome downregulate the expression of the tight junctions (TJs) [91], exposing both organs to biomacromolecules and microorganisms and triggering the neuroinflammation process [92].

The intestinal microbiome regulates the differentiation and maturations of innate immunocytes, such as macrophages, innate lymphoid cells and dendritic cells [93]. Highly specialized macrophages that are CNS tissue-resident constitute the microglia, which account for 5–15% of total brain cells. The microglia plays an important role in neurogenesis and the shaping of neuronal circuits, having implications for the further development of cognitive functions and social behavior [94]. During later developmental stages and adulthood, this tissue adopts a predominantly immune function activating either pro- or anti-inflammatory signaling cascades depending on the nature of the insult. Microglia from mice with a limited microbial complexity displayed genetic and morphological features similar to the microglia observed in GF mice. These alterations were reversed by recolonization of the gut microbiome through 6-week cohabitation of GF mice with control mice [95]. Therefore, a constant input from the gut microbiome is required by the microglia to adequately fulfill their role in neuronal maturation and immune surveillance [96]. Additionally, the gut microbiome influences the recruitment of ‘trafficking’ monocytes from the periphery to the brain [97]. This recruitment seems to be mediated by the TNF-α and reversed by the administration of probiotics in preclinical studies [98]. The free fatty acid receptors 2 (FFAR 2), which are G protein-coupled receptors located in peripheral lymphocites, may also be involved in this trafficking [99]. This would provide additional support for the implication of the gut microbiome, as SCFAs are natural ligands for FFARs [100].

Acquired immunity develops and matures during exposure to the gut microbiota. Intraepithelial lymphocytes (CD4/CD8 double-positive alpha-alpha T cells specifically) were reported to require the presence of *L. reuteri*, together with a tryptophan-rich diet, to reprogram intraepithelial CD4-positive T cells into immunoregulatory T cells [101]. In a study exploring the effects of chronic stress on long-lasting altered levels of IL-10+ T regulatory cells, an association was found between the concentrations of IL-10 and the abundance of *Clostridium* [102]. Absence of microbiota also reduces the content of immunoglobulins A and G1 (IgA and IgG1) and increases the levels of immunoglobulin E (IgE), thereby inducing the appearance of diverse diseases [12]. A recent study determined the microbial composition and immunoglobulin profile in fecal samples from 32 healthy infants with a high risk of developing type 1 diabetes as determined by human leukocyte antigen (HLA) genotyping. IgA levels correlated with relative abundances of *Bifidobacteria* and Enterobacteriaceae, whereas IgG levels were associated with *Haemophilus* [103].

## 3. Microbiota–Gut–Brain Axis and Attention-Deficit and/or Hyperactivity Disorder (ADHD)

### 3.1. ADHD: Clinical and Pathophysiological Aspects

ADHD is the most common neurodevelopmental disorder in children and adolescents, affecting 5% of individuals younger than 18 years [104,105]. It is characterized by the permanent and impairing presence of inattention and/or hyperactivity and impulsivity. These core symptoms must appear before the age of 12 in accordance with the new diagnostic criteria (DSM-5) [106]. The course of this disorder is variable, and some symptoms may persist into adulthood in around 40–60% of cases [107]. ADHD impacts on many aspects of an individual’s wellbeing, including physical health and academic, social and occupational functioning. It is frequently comorbid with other psychiatric and neurological conditions, such as ASD, mood disorders, epilepsy or sleep problems, creating a substantial burden for the individual, their family and the community [108]. Psychostimulants, and methylphenidate (MPH) in particular, represent the first-line medication for moderate and severe cases of ADHD in children from 5 years and over and in young patients [109]. Its efficacy mainly lies in increasing the extracellular levels of DA and norepinephrine (NE) [110], although it has additional effects on other neurotransmitter systems also involved in ADHD pathophysiology, such as 5-HT [111] and even Glu [112]. However, the long-term use of psychostimulants is often limited by poor compliance and tolerability problems derived from the combination of adverse effects, ADHD-related stigma and social resistance to medication, particularly in adolescents [113,114].

Numerous etiological factors have been attributed to ADHD: genetic factors, which represent around 70–80% [115]; and diverse environmental factors, including perinatal factors (prematurity, low birthweight) and psychosocial determinants (adoption, child neglect) [116,117,118]. The pathophysiology of ADHD is yet to be clarified. ADHD symptoms are associated with deficits in executive functions, such as behavioral inhibition, working memory, set-shifting, planning and organization [119]. The neuroanatomical basis for this impairment has traditionally been located in the prefrontal cortex [120,121]. However, several large neural networks have also been implicated in ADHD recently, particularly the dopaminergic mesolimbic system, which is associated with motivated behaviors, anticipated outcomes and reinforced learning [108,119]. In fact, it has been suggested that the alteration of the catecholaminergic neurotransmission system could be the main pathophysiological factor for ADHD [122,123].

Nowadays, there is increasing evidence that the aforementioned etiological factors and catecholaminergic dysfunction may lead to a neuronal state predominantly characterized by oxidative stress and inflammation, which could perpetuate the neurochemical alterations responsible for ADHD [124]. Increased levels of oxidative and nitrosative (NO) stress markers, together with a decrease in the concentrations of antioxidants, have been found in ADHD [125,126]. Furthermore, an alteration in the mitochondrial number and function in the dopaminergic neurons have been reported in individuals with ADHD in comparison with controls [127,128]. A dysregulation of the mitochondrial function provokes an uncontrolled production of reactive oxygen species (ROS) and reactive oxygen nitrogen species (RONS), which are by-products of the oxidative reactions leading to the production of adenosine triphosphate (ATP) [129]. Excessive levels of ROS/RONS harm the integrity of neurons by oxidating the polyunsaturated fatty acids (PUFAs) that constitute their membranes, as well as alter the apoptotic mechanisms. ROS/RONS also prompt the activation of the microglia and the release of inflammatory cytokines and the nucleotide-binding and oligomerization domain (NOD)-like receptor family, pyrin domain containing 3 (NLRP3) inflammasome, creating a vicious cycle [130,131]. In fact, Oades et al. [132] reported elevated levels of the inflammatory interleukins IL-16 and IL-13 in children with ADHD, which would respectively be associated with hyperactive-impulsive symptoms and inattention.

### 3.2. Differential Gut Microbial Profiles in Patients with ADHD: Association with Symptoms and Pathophysiological Implications

Aarts et al. [133] were the first authors to report microbial composition differences in Dutch young adult patients with ADHD using the next-generation sequencing of 16S rDNA in fecal samples. No significant differences in either alpha diversity (within-sample), which accounts for species richness, or beta diversity (between-sample), which indicates differences in diversity between the two cohorts, were found between ADHD patients and healthy controls. However, within the phylum Actinobacteria, the genus *Bifidobacterium* was significantly increased in the ADHD cohort. The authors also predicted bacterial gene function in relation to the metabolic pathways involved in the synthesis of phenylalanine, tyrosine and tryptophan. Interestingly, the relative abundance in the genus *Bifidobacterium* was correlated with a significant increase in the enzyme cyclohexadienyl dehydratase (CDT), which is involved in the synthesis of a dopaminergic precursor (phenylalanine). In a subset of 28 participants, independent of diagnosis, Aarts et al. also performed functional magnetic resonance imaging (fMRI) analysis to correlate the differences in microbial composition with neural reward responses. They observed a negative association between the relative abundance of CDT and reward anticipation responses in bilateral ventral striatum. Reward anticipation, which is dependent on DA neurotransmission [134], is crucial to direct actions towards positively balanced stimuli and has been reported to be reduced in ADHD patients [135]. Aarts et al. highlighted that the differential microbiome composition found between patients with ADHD and controls in their study may account for altered reward anticipation responses, which is a neural hallmark of ADHD. The novelty of this study resided in it representing the first report on a genetic capacity of the gut microbiome to impact on the dopaminergic metabolic pathways in patients with ADHD. However, the results should be interpreted with caution due to several limitations: first, the age gap between controls and cases (27.1 vs. 19.5 years in average), as well as the differences in sample sizes (controls = 77 participants; ADHD subjects = 17 patients); second, the intake of medications by ADHD patients was not reported in detail; third, differences between groups in relation to potential confounders, such as dietary patterns or the intake of antibiotics, which could have influenced the differences found, were not mentioned either; finally, the inclusion of unaffected ADHD patients’ siblings in the could have affected the representativeness of the control group. The gut microbiome of siblings to ADHD patients could express an intermediate phenotype between patients and non-related controls. Therefore, it may not represent an appropriate reference group for comparison [136].

In order to avoid the potential interference of ADHD medication, Jiang et al. [137] analyzed the microbial composition of a group of 51 treatment-naïve ADHD children, and compared it with a cohort of 32 healthy controls. No significant differences in alpha or beta diversity were found between groups, but a significantly lower concentration of the genus *Faecalibacterium* (from the family Ruminococcaceae) was reported in the ADHD group. These authors also found a negative association between the abundance of *Faecalibacterium* and parental reports of ADHD symptoms. Interestingly, low levels of *Faecalibacterium* have been reported in atopic diseases such as asthma, eczema and allergic rhinitis [138], which seem to be independently associated with ADHD [139]. Jiang et al. explained that the differences between their results and those obtained by Aarts et al. [133] could be due to differences in the age of the patients and the dietary pattern: the Chinese diet versus the typical high-fat Western dietary pattern. The main limitation of the study conducted by Jiang et al. was precisely its cross-sectional design, which prevented the authors inferring any conclusions about causality. By contrast, the methodology of this study was robust in terms of the selection and comparability between study samples: no significant differences in terms of age, gender, body mass index (BMI) or perinatal factors were found between patients and controls; the authors stratified for possible confounding factors, such as the use of previous use of probiotics/antibiotics, the current use of ADHD medication, the presence of atopic diseases and gastrointestinal, depressive or anxiety symptoms.

A recent study, conducted by Szopinska-Tokov et al. [140] on a Dutch sample of 42 adolescents and young adults with ADHD, revealed a significant increase of a genus from the family Ruminococcaceae. However, in this case the genus was *Ruminococcaceae_UGC_004*, which was found to be associated with inattention symptoms. This association was not affected by the intake of ADHD medication. Szopinska-Tokov et al. compared the ADHD population with other two cohorts of patients: subthreshold ADHD group, composed of individuals who did not reach the criteria for ADHD but scored too high to be considered healthy controls; and the control group, which included unaffected siblings of ADHD patients, although the family relatedness was a factor considered in the statistical analysis. The alpha diversity was not significantly different between groups, although the beta diversity was reported to be significantly reduced among patients with ADHD, and this correlated with inattention scores. The Basic Local Alignment Search Tool (BLAST) available at the National Center for Biotechnology Information (NCBI) was used to search for similarities between protein and nucleotide regions. Consequently, the authors found that the genus *Ruminococcaceae_UGC_004* shared sequences with microbial species with the ability to consume the GABA neurotransmitter. The novelty of this study was to provide new evidence on the role of the gut microbiome in neurotransmitter systems related to ADHD pathophysiology, via similarity between biological sequences. However, the main limitations of Szopinska-Tokov et al.’s study were related to the selection and comparability between samples: no information in relation to the recruitment process of the control and subthreshold ADHD samples was reported; information on lifestyle and dietary patterns, which may have interfered with the results, was not collected by the authors; intake of ADHD medication was registered through self-reports or parental reports on the day of measurement, which could have introduced a recall bias.

Prehn-Kristensen et al. [141] analyzed the differences in the microbial composition between a German population of 14 adolescents with ADHD and a cohort of 17 non-related controls. No differences in dietary intake were found between the two cohorts. A next-generation sequencing of 16S rDNA in fecal samples was performed. Alpha diversity was significantly reduced in patients with ADHD, and negatively correlated with the levels of hyperactivity. Beta diversity was also significantly different between the two cohorts, due to distinct abundances of different microbial taxa: at the genus level, *Prevotella* and *Parabacteroides* were detected as markers for the control group, whereas *Neisseria* was identified as marker for the ADHD group. At the family level, a significantly higher abundance of Bacteroidaceae was found in ADHD samples in comparison with controls. Interestingly, the reduction in alpha diversity was also observed in the mothers of ADHD patients, but not between the fathers of ADHD patients and controls. This finding suggests that alterations in the microbiome composition might be passed on maternally to the children. However, the study of Prehn-Kristensen et al. had several limitations: first, the small sample size; second, as no female patients were included in this study, it was not possible to explore if sex could influence the differences found; third, most ADHD patients were on medication, which was interrupted 48 h before being tested. This, however, might not have allowed enough time to avoid the potential impact of medication on microbial patterns [142]. The originality of this study was to provide some evidence in relation to the vertical transmission of microbial features. On the other hand, the overall quality of this study was reduced by the lack of representativeness of the samples, as all patients were males and no detailed description was provided in relation to the sources from which cases and controls were enrolled.

In a Chinese pediatric population, Wan et al. [143] compared the microbial composition of fecal samples between 17 patients with ADHD and 17 non-related healthy controls using shotgun metagenomics sequencing. No significant differences in alpha diversity were found between the two cohorts. At the genus level, *Faecalibacterium* (family *Ruminococcaceae*) was significantly decreased in ADHD patients, whereas the concentrations of *Odoribacter* (order *Bacteroidales*) and *Enterococcus* were significantly higher. At the species level, *Bacteroides caccae*, *Odoribacter splanchnicus*, *Paraprevotella xylaniphila* and *Veillonella parvula* were significantly increased in the ADHD group. The authors also analyzed the metabolic pathways associated with the microbial genes that were significantly different between the two samples of patients. Alterations in genes encoding enzymes involved in the dopaminergic synaptic pathways were found in the ADHD group. *Faecalibacterium* may exert anti-inflammatory effects, and their abnormal levels may lead to a higher expression of inflammatory factors that could contribute to ADHD pathogenesis [144,145]. *Enterococcus* has been reported to be associated with neurotransmitter release. It could lead to excessive intestinal conversion of levodopa into DA. However, peripheral DA is not able to cross the BBB to enter the CNS, therefore reducing the effectiveness of levodopa [146]. Additionally, significant higher levels of *Enterococcus* have been found in mice lacking the 5-HT transporter, which can lead to decreased 5-HT concentrations [147], a neurotransmitter also involved in the pathophysiology of ADHD. The higher concentrations of *Odoribacter* found by Wan et al. in ADHD patients were in line with a previous study that found increased levels of this genus in pediatric acute-onset neuropsychiatric syndrome (PAN) and pediatric autoimmune neuropsychiatric disorders associated with streptococcal infections (PANDAS) [148]. The main limitations of this study were the small sample size and the lack of information on the intake of ADHD medication. Additionally, the authors did not establish associations between ADHD core symptoms and microbial composition. However, the quality of this study is supported by its robust methodology: detailed description of the recruitment procedure; stratification for confounding factors in both samples (probiotics, allergic diseases, digestive or respiratory symptoms, dietary habits); and the use of whole genome shot gun sequencing (WGS), which provides a much more reliable estimation of the functional potential of the microbiome in comparison with the sequencing of 16S rDNA subunits of highly conserved genetic sequences [149].

Casas et al. [150] conducted a case-cohort study to investigate the influence of indoor microbial diversity early in life on the development of hyperactivity/inattention symptoms. Patients (*n* = 226) were selected from the Influence of Life-style factors on the development of the Immune System and Allergies in East and West Germany (LISA) birth cohort. This was a population study in which healthy full-term neonates were recruited from different hospitals in Germany and their bedrooms’ floor dust samples were collected at the age of 3 months. Indoor bacterial and fungal diversity were respectively analyzed by 16S rDNA gene sequencing and high-throughput sequencing. Fungal and bacterial alpha diversity metrics were calculated (richness, Shannon, Simpson and Chao1 diversity indices). The Strength and Difficulty Questionnaire (SDQ) was used to evaluate hyperactivity/inattention behavior at the ages of 10 and 15 years. A total of 23 children at the age of 10 years reached criteria for ADHD, while this number increased to 50 individuals at the age of 15 years. At the age of 10, bacterial richness (number of different taxa) was found to be inversely correlated with ADHD prevalence, whereas the number of fungal species was positively associated with a high prevalence. However, at the age of 15, only the Shannon index was significantly associated with hyperactivity/inattention symptoms, directly for bacteria and inversely for fungi. Therefore, this study suggests that early life microbial environment may be associated with the development of behavioral problems during childhood. However, the direction of the associations observed was heterogeneous and the observational design of the study did not allow establishing causality. Furthermore, the authors did not consider other confounding factors which could have influenced the results, such as certain lifestyles or the exposure to phthalates. With the collection of dust samples at one single time-point (3 months of age), changes in the indoor microbial environment across time were not considered either. An additional limitation of the study conducted by Casas et al. was in relation to the representativeness of the sample: the assessment of inattention/hyperactivity symptoms was only based on scores obtained in the parent-completed and self-completed Strengths and Difficulties Questionnaire (SDQ), which could have introduced a reporting bias.

In a double-blind placebo-controlled trial carried out by Stevens et al. [151], 17 male children with ADHD from New Zealand were randomized to take capsules containing placebo (*n* = 7) or a blend of vitamins, minerals and antioxidants (n = 10) for a period of 10 weeks. The aim was to investigate the effect of micronutrient supplementation on the human fecal microbiome composition. Sequencing of 16S rDNA was performed on fecal samples collected at baseline and after supplementation. Micronutrient supplementation was associated with 50% responder rates vs. 29% responder rates in the placebo group. The response resulted in enhanced overall function, improved attention, emotional regulation and aggression. However, no significant differences in the scores obtained in either the Children’s Global Assessment Scale (CGAS) or the ADHD Rating Scale-IV (ADHD-RS-IV) were found between the treatment and the placebo group. There were no significant changes in the alpha or beta diversity between the placebo and the micronutrient group, although patients receiving micronutrients showed increased community richness after 10 weeks supplementation in comparison with placebo. Micronutrient supplementation caused a significant decrease of the phylum Actinobacterium, particularly a 25% reduction in the order *Bifidobacteriales*, which was attributed to the genus *Bifidobacterium*. This was accompanied by higher levels of the genus *Collinsella*. A pairwise correlation was detected between lower ADHD-RS-IV scores and decreased Actinobacteria abundance, as well as between lower concentrations of Actinobacteria and higher scores in the CGAS (in which higher results indicate better functioning). The results highlight a potential effect of micronutrient supplementation to modulate the abundance of putative probiotic bacterial species. Nevertheless, the role of *Bifidobacterium* in ADHD is contradictory. Several studies reported a protective effect of *Bifidobacterium longus* on several neuropsychiatric disorders, including ADHD [136,152]. On the contrary, Aarts et al. [133] found an association between higher abundances of Bifidobacterium species and ADHD. These opposing results may account for differences between studies in sample sizes, dietary patterns, diagnostic heterogeneity of neuropsychiatric disorders and the interpretation of compositional datasets in general. The strengths of this study lie in the quality of its methodology: detailed description of the recruitment procedure and of clinical and analytical assessments performed to participants; stratification for confounding factors, such as dietary patterns and nutritional deficiencies. However, caution should be exercised when interpreting the results, as this was a pilot study with a small sample that only included male participants. Information on the intake on medication was not reported by the authors either.

In a Taiwanese study, Wang et al. [153] compared fecal microbiota compositions and dietary patterns between 30 naïve-medication children with ADHD and a cohort of 30 healthy controls. Dietary habits were explored with the use of a food frequency questionnaire including 49 food items from eight food groups. Microbial profiles were identified through 16S rDNA sequencing. Alpha diversity was assessed based on the Shannon, Chao1 and Simpson indices. Children with ADHD showed significantly higher values of Shannon and Chao 1 indices, although the Simpson index was significantly lower compared to controls. No differences in beta diversity were observed between the both groups. At the genus level, the microbial profile was fairly similar between patients and controls. Performing a linear discriminant analysis effect size (LEfSe), the relative abundance of *Bacteroides coprocola* was significantly lower in the ADHD group, whereas *Bacteroides uniformis*, *Bacteroides ovatus* and *Sutterella stercoricannis* were significantly increased. The genus *Fusobacterium* was relatively enriched in ADHD patients, while the relative abundance of *Lactobacillus* was higher in control subjects. Additionally, dietary patterns differed between the two cohorts of participants. Participants with ADHD showed a higher intake of refined grains and a lower proportion of vitamin B2 and dairy. The amount of *S. stercoricannis* was correlated with the intake of dairy, nuts/seeds/legumes, ferritin and magnesium. *B. uniformis* was associated with fat and carbohydrate intake, whereas no correlations were observed between *B. ovatus* and *B. coprocola* with any of the items included in the food frequency questionnaire. A positive correlation was found between ADHD symptoms and both *S. stercoricannis* and *B. ovatus*. The sample selection process was carefully described by the authors, who also considered some confounding factors, such as the use of probiotics, antibiotics, special dietary patterns and the presence of neuropsychiatric comorbidities. However, the main limitations of this study were found in relation to the representativeness and comparability of samples: ADHD patients with no neuropsychiatric comorbidities who scored lower than typical ADHD patients in the Swanson, Nolan, and Pelham Version IV Scale (SNAP-IV); and cases and controls who significantly differed in their dietary habits. These differences in the dietary pattern could have been the only reason for distinct microbial profiles.

Cheng et al. [154] applied gene set enrichment analysis (GSEA) to explore potential relationships between the gut microbiome and five different neuropsychiatric disorders: ADHD, ASD, bipolar disorder, schizophrenia and major depressive disorder. GSEA is capable of identifying groups of genes that share common biological functions or target at common diseases. Data were collected from publicly available genome-wide association studies (GWAS) from the Psychiatric GWAS Consortium. The ADHD genomic dataset comprised 19,099 patients with ADHD and 34,149 controls. The GSEA algorithm determined which ADHD-related genes or single nucleotide polymorphisms were enriched in published datasets from GWAS of the human gut microbiome. The genus *Desulfovibrio* and the order Clostridiales were significantly associated with ADHD. However, *Desulfovibrio* was also associated with ASD, which suggested that the abundance of this genus was not specific of ADHD. The novelty of this study consists in screening candidate gut microbiome for subsequent functional studies investigating the role of the gut microbiota in the pathophysiology of diverse neuropsychiatric disorders. The authors also used DNA level GWAS data, so that the results were less likely to be affected by environmental and dietetic factors. However, an important limitation of this study was that the gut microbiome-related gene sets were collected from previously published GWAS of gut microbiome, which are still quite scarce. Therefore, only a limited number of datasets was available for analysis.

The main findings and particular aspects of the aforementioned studies are summarized in Table 1.

The previous evidence argues for a different microbial composition in patients with ADHD. However, the results are too heterogeneous to draw confident conclusions regarding whether a specific microbial profile is associated with ADHD. Several factors may have contributed to the differences found between studies: diverse geographic, cultural, demographic and dietary characteristics of study populations; differences in relation to the selection of control groups (siblings or non-related controls) or the intake of ADHD medication; methodological differences associated with the sampling and storing, microbiome sequencing and the choice of bioinformatics pipeline and reference databases [155,156].

Notwithstanding heterogeneity, we attempted to group studies according to the differential microbial features in patients with ADHD:The results obtained by Stevens et al. [151] agreed with those previously found by Aarts et al. [133]. According to Stevens et al., supplementation with micronutrients reduced the abundance of the genus *Bifidobacterium*, which belongs to the phylum Actinobacterium. The improvement of ADHD symptoms after supplementation could have been influenced by these microbial changes, as a correlation was found between a reduction in ADHD-RS-IV scores and a decreased abundance of Actinobacteria. Previously, Aarts et al. had showed a significant increase of the genus *Bifidobacterium* in young patients with ADHD, establishing a correlation between this genus and a particular DA-related enzyme (CDT) associated with altered neural reward responses. In both studies, no differences in alpha/beta diversities were found between patients and controls.Prehn-Kristensen et al. [141] revealed a significantly higher abundance of the family *Bacteroidaceae* in adolescents with ADHD. This was corroborated later by Wang et al. [153], who found an increase of some species of *Bacteroides* in the ADHD group. However, the results obtained in relation to alpha and beta diversity were discordant: alpha diversity was reduced and significant differences in beta diversity were found in the ADHD group in the study carried out by Prehn-Kristensen et al. By contrast, no significant differences in beta diversity between groups were found by Wang et al., who also highlighted higher values of Shannon and Chao1 indices in the ADHD group. This could account for methodological differences between studies: sample size, gender, intake of ADHD medication, ethnicity, and dietary habits.Cheng et al. [154] showed significant associations between the order *Clostridiales* and ADHD. Interestingly, other three studies are in line with this finding: a significant increase of the genus *Ruminococcaceae_UGC_004* (family Ruminococcaceae, order Clostridiales) in the ADHD group was shown by the study conducted by Szopinska-Tokov et al. [140]. Jiang et al. [137] and Wan et al. [143] found a significant decrease of the genus *Faecalibacterium* in patients with ADHD, which belongs to the same family. In any case, these results are not mutually exclusive, as the increase in a genus can be accompanied by a lower abundance of a different genus from the same family. For instance, in the study performed by Stevens et al. [151], the decrease of the genus *Bifidobacterium* occurred along with an increase of the genus *Collinsella*, from the same family study. Differences in alpha and beta diversity between studies could again be attributed to demographic and clinical differences between participants (medication status, dietary habits), and divergences in sampling and storing.

According to the previous studies, it seems clear that the gut microbiome could play a role in the pathophysiological mechanisms leading to ADHD symptoms. As explained before, the gut microbiome may interfere in the catecholaminergic neurotransmission system by either influencing their metabolic pathways or the expression of the genes encoding these neurotransmitter transporters [147]. Furthermore, the gut microbiome may contribute to exacerbate the mechanisms of neuroinflammation and oxidative stress that are present in ADHD. This is not only due to their impact on the microglia and the BBB permeability, but also to the production of SCFAs. As mitochondrial fuels, they may contribute to the uncontrolled production of ROS/RONS, particularly when the mitochondrial function is already altered [157] and the synthesis of SCFAs is increased due to alterations in the microbial composition. In fact, some bacterial species are more active in the production of SCFAs, such as *Bacteroides* spp. and *Clostridiae* spp. [158]. SCFAs could even have additional effects on neurogenesis, as they could influence the levels of brain-derived neurotrophic factor (BDNF) [45]. Changes in the serum concentrations of BDNF have been described in patients with ADHD [159], which were regulated after MPH treatment [160]. An additional pathway that could underpin the role of the gut microbiome in ADHD pathophysiology is its interaction with the metabolism of omega-3 PUFAs, although this will be described in the next section.

A schematic representation of bacterial taxonomy to facilitate comprehension is shown in Figure 2.

### 3.3. Gut Microbiome and Omega-3 Polyunsaturated Fatty Acids in ADHD

Omega-3 PUFAs, and particularly docosahexaenoic acid (DHA) and eicosapentaenoic acid (EPA), play an important role in membrane fluidity, neurotransmission and receptor function [163,164]. They affect the levels of BDNF and glial cell-derived neurotrophic factor (GDNF), a neuroprotective and important trophic factor in dopaminergic neurons [165]. Lower brain DHA content during development is associated with frontocortical dopaminergic hypofunction [166]. In animal male models of ADHD, a diet enriched in omega-3 PUFAs was followed by an increased striatal turnover of DA, 5-HT, improved attention and decreased impulsivity [167]. Omega-3 PUFAs also have essential anti-inflammatory properties, as they reduce the levels of the pro-inflammatory interleukin IL-1β by inhibiting the activation of the NLRP3 inflammasome [168].

Total omega-3 PUFAs are significantly decreased in pediatric patients with ADHD, who showed a significant higher omega-6:omega-3 fatty acids ratio in comparison with controls [169]. Omega-3 PUFAs are a focus of increasing interest in children with ADHD nowadays, as they might represent an alternative or adjuvant therapy to psychostimulants in ADHD. Although studies on supplementation with omega-3 PUFAs in children and adolescents with ADHD have shown mixed results, DHA/EPA might contribute to an improvement in total symptom score, inattention and hyperactivity [170,171].

There is growing evidence to support a bidirectional relationship between omega-3 PUFAs and the gut microbiome. In mice, an 8-week dietary supplementation with different strains of *Bifidobacterium breve* (*NCIMB 702258* and *DPC 6330*) yielded different fatty acid profiles in the host tissues. Mice fed with *B. breve NCIMB 702258* showed significantly higher concentrations of arachidonic acid (AA) and DHA in the brain in comparison with those fed with *B. breve DPC 6330* and the non-supplemented control group. Nevertheless, both groups of *B. breve*-supplemented mice presented significantly lower brain concentrations of the omega-6 dihomo-γ-linolenic acid (20:3 n-6). The results were not due to differences in dietary patterns others than the supplementation received. They were also accompanied by changes in the gut microbial composition: at the family level, there was a higher abundance of Clostridiaceae and lower concentrations of *Eubacterium* in the supplemented groups, particularly in those receiving *B. breve DPC 6330* [172].

On the other hand, omega-3 PUFAs also seem to influence the gut microbial composition. A group of pregnant female mice and their male offspring were fed with a control diet, omega-3 (DHA + EPA) enriched diet or omega-3 deficient diet. All three diets only differed in the fatty acid content. Cognitive performance and social behavior were assessed in male offspring at adolescence and adulthood, in parallel with the microbial profile, analyzed through 16S rDNA sequencing. Animals receiving the omega-3 deficient diet showed impaired communication, social and depression-related behaviors, whereas the mice fed with omega-3 enriched diet displayed improved cognition. On a microbial level, a significant increase in the Firmicutes:Bacteroidetes ratio was observed in the mice following a diet deficient in omega-3 fatty acids. At the genus level, mice following a diet abundant in omega-3 showed a significant increase in *Lactobacillus* and *Bifidobacterium* during adulthood in comparison with the other two groups [173]. In a recent study, male rats were randomized to receive a control diet in non-stressing conditions (NSCD), a control diet in social instability stress conditions (SCD), and an omega-3 (DHA + EPA) and vitamin A enriched diet during stressful manipulation (SED). The aim of this study was to explore if omega-3 PUFAs + vitamin A could prevent the behavioral deficits and microbial changes induced by stressing circumstances during adolescence. Cognitive performance in the SED group was indistinguishable from that of the NSCD group. BDNF expression in the hippocampus was decreased in SCD rats in comparison with NSCD, and this change was prevented by the enriched diet. The enriched diet significantly increased alpha diversity and prevented the changes in the microbial composition induced by stress. A significant increase in the production of unbranched SCFAs was also observed in the SED group [174]. These results should be interpreted cautiously, considering that they were performed in small animal populations during strict laboratory conditions. Furthermore, no associations between animal behavioral features and changes in the microbial composition were performed by the authors.

Similarly, in a randomized, open-label, cross-over trial of 8-week supplementation with omega-3 PUFAs (DHA/EPA) to 22 healthy volunteers, Watson et al. [175] reported an increase in the butyrate-producing genera *Bifidobacterium*, *Roseburia* and *Lactobacillus* in fecal samples. No significant changes in alpha or beta diversity were found after supplementation. However, the authors found no correlation between the microbial changes and the concentrations of EPA/DHA measured in red blood cells after supplementation. The recruitment process was carefully described by the authors, who also stratified for confounding factors such as the current use of omega-3 PUFA supplements, concomitant use of non-steroidal anti-inflammatory medication, previous surgery for bowel resection, current treatment for any chronic inflammatory condition or malignancy, pregnancy or smoking. As limitations, this study included a small sample size and no information on dietary habits was recorded.

A representation of the interaction between the gut microbiome and omega-3 PUFAs, and how this relationship could interfere with ADHD pathophysiological mechanisms, is shown in Figure 3.

### 3.4. Gut Microbiome and ADHD Comorbidities: Chronodisruption and Sleep Disorders

Sleep disorders represent one of the most frequent comorbidities in children with ADHD. They can be present in up to 70% of patients [176], affecting their cognitive, behavioral and physical state [177], and thus increasing parental stress levels [178]. Patients with ADHD suffer from variable sleep disruptions, but the most consistent finding is probably the presence of a delayed circadian phase (evening preference), with a consequent disturbance in daytime functioning [179,180]. The exact neurobiological mechanisms underlying the sleep disorders in ADHD are yet to be fully determined, but they have been associated with a circadian dysfunction in which the evening increase in endogenous melatonin secretion (dim light melatonin onset) is significantly delayed [181,182]. Pineal melatonin synthesis is controlled by the central circadian clock in the CNS, which generates circadian rhythms through the transcriptional/translational feedback loops existent between the different clock genes (*Bmal:Clock*, *Per:Cry*, *Rorα*, *Rev-erbα*, *Chrono*) [129]. Indeed, several polymorphisms of the CLOCK gene might be associated with susceptibility to ADHD [183,184]. 

The diversity and composition of the gut microbiome oscillate during the 24 h light-dark cycle, suggesting that the gut, as the rest of the tissues, has its own peripheral circadian clocks, which ultimately depend on the central clock. In animal studies, the diversity and abundance of certain species, such as *Bacteroidetes* and *Clostridia*, swayed during the light-dark period [185]. The number of bacteria was higher during the active phase of mice, with a higher abundance of *Bacteroidetes*, whereas bacterial load was reduced during the rest phase, with a predominance of the phylum *Firmicutes* [186,187]. Knockout of clock genes, including *Per1/2*, attenuated these oscillations [185,188]. Additionally, melatonin seems to impact on the richness and diversity of the intestinal microbiota and the *Firmicutes:Bacteroidetes* ratio in mice [189]. Recently, Gao et al. [190] evaluated the changes induced by melatonin treatment in the microbial composition and the colonic mucosal integrity of sleep-deprived mice, in which plasma melatonin concentrations were lower than in controls. Sleep deprivation was associated with a significant decrease in microbial diversity and richness, and a significant increase in the Firmicutes:Bacteroidetes ratio, in comparison with control mice. At the genus level, a significant decrease of *Akkermansia*, *Bacteroides* and *Faecalibacterium*, together with an increase of *Aeromonas*, was found in mice with sleep deprivation. Goblet cells and the expression of TJ proteins were also significantly reduced by sleep deprivation. By contrast, melatonin administration restored the richness and diversity indices, as well as the *Firmicutes:Bacteroidetes* ratio, to values similar to the control group. After treatment with melatonin, a significant decrease of *Aeromonas* was observed, along with a rise in the content of *Akkermansia*, *Bacteroides* and *Faecalibacterium*. Furthermore, melatonin significantly increased the number of goblet cells and the expression of TJ proteins. These changes were in parallel with a neutralization of the disrupting effects provoked by sleep deprivation in the balance between pro-inflammatory/anti-inflammatory cytokines and the redox status.

In a Chinese population of 120 children with ASD, Hua et al. [191] explored the differences in the composition of the gut microbiome and its metabolites between those patients who suffered from sleep disorders (*n* = 60) and those without sleep problems (*n* = 60). Sleep disorders were assessed through the Children Sleep Habits Questionnaire (CSHQ). Microbial DNA was extracted from stool samples and analyzed through next-generation sequencing of 16S rDNA. No information on medication or dietary patterns was included. Abundance and richness of the gut microbiota were significantly higher in ASD children with sleep problems. No significant differences were observed in the Firmicutes:Bacteroides ratio, but at the genus level, a significant reduction in the butyrate-producing bacteria *Faecalibacterium* and *Agathobacter* was found in the sleep disorder group. The abundance of each of the two genera was negatively correlated with CSHQ scores. Three differential metabolites were observed between sleep disorder and no sleep disorder groups: concentrations of 3-hydroxybutyric acid (a butyrate-derived acid) and melatonin were significantly lower among patients with ASD and sleep problems, whereas levels of 5-HT were significantly higher. The concentrations of 3-hydroxybutyric acid were correlated with melatonin levels and *Faecalibacterium* abundance. Additionally, a positive correlation was also found between melatonin concentrations and the abundances of *Faecalibacterium* and *Agathobacter*. Due to the design of the study, it was not possible to elucidate if the differences observed in the gut microbiome were due to a circadian dysregulation in patients with ASD, or if a previous alteration in the microbial composition contributed to the changes observed in melatonin concentrations. Another limitation was the absence of a control group to be compared with the populations with ASD, apart from the absence of information on dietary patterns and medication intake. Notwithstanding, this study provided compelling evidence for an association between microbial metabolites (butyrate) and melatonin. Studies on the bidirectional relationship between the gut microbiome and the sleep/wake circadian rhythm in ADHD are lacking. However, given the evidence provided and the genetic overlap between ASD and ADHD [192], it seems plausible to suggest the existence of similar alterations in children with ADHD and sleep disorders.

### 3.5. Therapeutic Role of the Gut Microbiome in ADHD: Probiotics

Given the documented involvement of the MGBA in the pathophysiological mechanisms of ADHD, it may be reasonable to consider the gut microbiome as a potential therapeutic target for this disorder. In this regard, probiotics are living non-pathological microorganisms that provide a health benefit by improving physiological conditions in the host when they are administered in adequate amounts, well as a food ingredient, supplement or as a drug [193]. To date, species of the genus *Lactobacillus* and *Bifidobacterium* represent the most investigated strains [194].

A recent systematic review included randomized controlled trials, published between 1990 and 2018, that had evaluated the effects of probiotic supplementation on the cognitive function in children and adolescents. Seven studies were found to meet the inclusion criteria, but only one of them reported a significant reduction in the risk to develop ADHD or ASD [195]. This was a double-blind randomized placebo-controlled trial conducted by Pärtty et al. [152] on a Finnish population of 159 infants who had at least one family member with allergic disease. Their mothers had been randomized to receive a daily supplementation with *Lactobacillus rhamnosus GG* (1 × 10^10^ colony-forming units (CFU)) or placebo, starting 4 weeks before expected delivery. The intervention was continued for 6 months after delivery, given either to the children or to the mother, if they were breastfed. Behavioral patterns and fecal microbiome composition of the children were periodically analyzed until 13 years of age. The diagnosis of ADHD or ASD was established by an experienced child psychiatrist or neurologist in accordance with the International Classification of Diseases 10th Edition (ICD-10). Techniques of fluorescein in situ hybridization (FISH) and quantitative polymerase chain reaction (qPCR) were used to analyze the gut microbiome. Seventy-five subjects completed the study, 40 in the intervention group, and 35 participants in the placebo group. Significant differences were found in the percentage of patients who were diagnosed with ADHD or ASD at the age of 13: 17.1% children in the placebo group vs. none in the probiotic group. At the genus level, the content of *Bifidobacterium* at 6 months of life was significantly lower among children who later developed neuropsychiatric disorders, although this difference was not present at 13 years of age. All children diagnosed with neurodevelopmental disorders were male, but the differences found were not influenced by gender when controlling for this factor in a logistic regression analysis. Perinatal factors, such as delivery mode, birth/length weight and infant feeding mode were comparable between placebo and intervention groups. The recruitment process was detailed by the authors, who also used accurate techniques of FISH and qPCR. However, some limitations should be taken into account when interpreting these results: a considerable percentage of drop-outs (around 50%) may have biased the results; no information on maternal dietary habits during pregnancy and after delivery (in the case of breastfed infants) was recorded; additional information on risk factors for ADHD, such as maternal smoking, nutritional deficiencies or psychosocial factors was not reported either; finally, no association were established between the abundance of *Bifidobacterium* at 6 months of life and ADHD psychometric scores.

By contrast, less consistent results were shown by a more recent double-blind pilot randomized placebo-controlled trial on children and adolescents with ADHD who received a 3-month supplementation with *Lactobacillus rhamnosus GG* ATCC53103 (once-daily dose of 1 × 10^10^ CFU). Thirty-two naïve-medication patients with ADHD (4–17 years old) were randomized to take the probiotic (*n* = 18) or placebo capsules (*n* = 14). The serum profile of pro/anti-inflammatory cytokines was assessed at baseline and after 3 months, as well as the scores obtained in the following ratings: the ADHD Parent Report Rating Scale-IV, which is based on the diagnostic criteria for ADHD as established by the Diagnostic and Statistical Manual of Mental Disorders 4th Edition (DSM-IV); the Child Self-Report and Parent Proxy-Report of the Pediatric Quality of Life Inventory^TM^ (PedsQL^TM^) 4.0 Generic Core Scale; the Parent Form (CBCL/6-18) and the Teacher Report Form (TRF) of the Child Behavior Checklist (CBCL) of the Achenbach System of Empirically Based Assessment (ASEBA). The most interesting finding was a significant improvement in the scores obtained in the PedsQL^TM^ Child-Self Report after 3 months only in the probiotic group. This may reflect that these patients felt an improvement in physical, emotional and social terms, as well as in their school functioning. Mixed results were obtained in relation to the other tests: the PedsQL^TM^ Parent-Proxy Report showed significantly better scores only in the placebo group, whereas a significant improvement was reflected by the scores obtained in the ADHD Rating Scale and the CBCL Parent Form in both groups of participants. The differences observed in the serum levels of cytokines were also ambiguous: the proinflammatory IL-12 p70 and TNF-α significantly decreased in the probiotic group only, whereas significantly lower concentrations of the proinflammatory IL-6 was found in both groups; a significant reduction of the anti-inflammatory IL-10 was also observed, and only in the probiotic group. The authors attributed these controversial results to both the small sample size and the short observation period [196]. This pilot study had several limitations: the age range of patients in both groups was too wide (4 to 17 years old), which questioned their comparability in relation to the psychometric scores and the cytokine profile; no information in relation to dietary habits was included. Additionally, possible changes in the microbial profile were not measured either. Establishing associations between shifts in the microbial composition and changes in serum cytokine levels and psychometric scores would have provided more consistency to this study.

In a double-blind randomized placebo-controlled trial, Skott et al. [197] aimed at investigating the effects of a synbiotic on ADHD core symptoms, autistic comorbid symptoms and daily functioning in a population of children and adults with ADHD. A synbiotic is a product that combines prebiotics and probiotics and in which the prebiotic component favors the probiotic strains [198]. A prebiotic is a substrate that is selectively utilized by host microorganisms and confers a health benefit [199]. The synbiotic used in this study was Synbiotic 2000, a lyophilized composition of 4 × 10^11^ CFU per dose of three lactic acid bacteria (*Pediococcus pentosaceus*, *Lactobacillus casei* spp. *paracasei* and *Lactobacillus plantarum*) and the fermentable fibers betaglucan, inulin, pectin and resistant starch. The pediatric sample was composed of 68 children and adolescents with ADHD (10–14 years) who were allocated to take Synbiotic 2000 (*n* = 34) or placebo capsules (*n* = 34) for a period of 9 weeks. The following questionnaires were completed before and after intervention: the Swanson, Nolan and Pelham-IV scale (SNAP-IV) parent rating scale to measure ADHD symptoms; the Weiss Functional Impairment Rating Scale-parent-reported for child (WFIRS-PC) to assess functional impairment; and the Social Communication Questionnaire (SCQ) to evaluate autistic symptoms. Given that Synbiotic 2000 has documented anti-inflammatory properties [200,201], the authors also decided to measure blood levels of C-reactive protein (CRP) and vascular cell adhesion molecule-1 (VCAM-1) at baseline, as both of them are involved in the inflammatory response. No significant differences between groups (synbiotic vs. placebo) were observed for the changes found in the scores of ADHD symptoms (SNAP-IV) and functional impairment (WFIRS-PC) after intervention. Only a tendency for a reduction of autistic symptoms, according to the SCQ scores, was observed in the synbiotic group. However, when stratifying for VCAM-1 levels and the intake of ADHD medication, it was observed that the tendency of Synbiotic 2000 to reduce SCQ total score was driven by the existence of elevated VCAM-1 concentrations and the absence of ADHD medication. The novelty introduced by this clinical trial was to explore for the first time the effects of a synbiotic on symptoms and functioning in pediatric patients with ADHD. However, several limitations are highlighted for this study: first, a food frequency questionnaire was completed by participants only at baseline, thus it was unlikely to have included possible dynamic changes in dietary patterns over the 9-week intervention period; second, after stratifying for VCAM-1 levels and medication, the sample sizes were substantially diminished; third, CRP and VCAM-1 levels were only measured at baseline, so it was not possible to observe effects of the synbiotic on proinflammatory markers; fourth, patients were also taking melatonin, omega-3 PUFAs and other types of probiotics at baseline, which were not considered as confounding factors for the statistical analysis; fifth, no information on the intake of ADHD medication was reported.

An observational study on 2467 very low birth-weight (VLBW) infants who were followed until the age of 5–6 years, showed no association between probiotic treatment and neurocognitive outcome of the study population. However, breastfeeding for a minimum of 3 months was associated with lower scores of conduct disorder and inattention/hyperactivity. The probiotic formulation consisted of a standardized combination of *Bifidobacterium infantis* (1–1.5 × 10^9^ CFU) and *Lactobacillus acidophilus* (1–3 × 10^9^ CFU), administered once-daily from days 1–3 of life until days 14–35 of life. Neurocognitive outcome was assessed by considering the scores in the parent-reported Strength and Difficulties Questionnaire (SDQ), and the intelligence quotient obtained in the Wechsler Preschool and Primary Scale of Intelligence 3rd Edition (WPPSI-III) test. This represents the first large-scale study on neurocognitive outcome after probiotic administration during the neonatal period. The sample was carefully selected, stratifying for potential confounding factors, such as gestational age, motor impairment, maternal age, maternal educational level, multiple birth and bronchopulmonary dysplasia. However, the study exhibited several limitations that should be taken into account when interpreting the results: first, the design was observational, so it was not possible to draw causal conclusions based on data; second, breastfeeding was considered as either exclusive breastfeeding or mixed breastfeeding combining breast milk and infant formula; third, information on the duration of breastfeeding was collected by a parent-reported questionnaire at the age of 5 years, which may have introduced a recall bias; finally, a combination of biological and psychological factors that could have influenced neurodevelopment in the time lapse between perinatal period and 5–6 years of age were not included in the study [202].

In general, the aforementioned studies provide some evidence to suggest a therapeutic role of the gut microbiome in pediatric patients with ADHD. However, notable inconsistencies were found among the three supplementation clinical trials on children with ADHD. The most robust results were provided by Pärtty et al. [152], who reported a significant reduction in the proportion of participants who developed ADHD symptoms in the probiotic group. Interestingly, the abundance of *Bifidobacterium* at 6 months of age was significantly lower among those who developed neuropsychiatric disorders. This would contrast with the results obtained by Aarts et al. [133] and Stevens et al. [151], as previously mentioned in Section 3.2. Nonetheless, the relative abundance of *Bifidobacterium* referred by Pärtty et al. at 6 months was not present at 13 years of age, when the diagnosis was made. The study of Pärtty et al. was only slightly corroborated by the randomized placebo-controlled trial conducted by Kumperscak et al. [196]. They showed a significant improvement in the PedsQL^TM^ Child-Self Report scores after supplementation. This study was conducted on a clinical population of children and adolescents with ADHD using exactly the same probiotic as Pärtty et al., but for a shorter period. However, the results obtained in the serum cytokine profile and in ADHD-RS-IV scores, a psychometric scale specifically designed to detect ADHD core symptoms, were mixed. Similar mixed results were also offered by the clinical trial carried out by Skott et al. [197] on a pediatric population with ADHD. Only a tendency to significance was found for the reduction in autistic symptoms in the group randomized to the synbiotic, whereas the ADHD symptom score (SNAP-IV) decreased to a similar degree in the supplemented and placebo groups. No analysis of microbial composition was performed in the last two clinical trials, which prevented from finding more consistent results across studies.

As suggested by Cerdó et al. [203], there are still considerable divergences between studies in dosage, type of strain, intervention period, microbiome composition analysis, methods for neurological assessment, identification of potential confounding factors, study design and sample size. For standardization purposes, future research is needed to identify the most effective doses and combination of probiotics, as well as the minimum intervention period to observe clinically meaningful results in ADHD symptoms and associated comorbidities.

## 4. Limitations

In the present review, we have aimed to provide the current evidence on: the role of the gut microbiome in ADHD pathophysiology and possible association with symptoms; implication of the gut microbiome in the omega-3/omega-6 PUFA imbalance found in patients with ADHD; interaction between the gut microbiome and circadian rhythms; and the therapeutic role of probiotics in pediatric patients with ADHD. Certainly, we have carefully selected and described the most recent studies on the aforementioned research lines. However, this is a narrative review, as such, our work is essentially informative and has limited capacity to assess the robustness of the results provided by the current literature in these topics. We have tried to improve this by: highlighting the limitations of the different studies, making a judgement of their novelty, quality and reliability; grouping studies according to their results, in order to show controversies and inconsistencies among them; and including final remarks with suggestions on the aspects that require to be improved in future studies. On the other hand, we cannot discard a possible selection bias, given by the absence of a detailed search strategy designed a priori. Therefore, a systematic review, with a previous comprehensive plan to identify all relevant studies on the topic and the use of statistical analysis to measure effect sizes, would have provided more solid conclusions to our initial hypothesis. A systematic assessment of the quality of the studies could also have been included, through the use of the Newcastle–Ottawa Quality Assessment scale for Cohort and Case-Control Studies (NOS) [204] and the Consolidated Standards of Reporting Trials (CONSORT Statement) [205].

## 5. Concluding Remarks

The gut microbiome maintains a bidirectional relationship with its host through neurological, hormonal and immune mechanisms, configuring the microbiome–gut–brain axis (MGBA). Alterations of the MGBA are responsible for the appearance of diverse neurological and neurodevelopmental disorders in pediatric populations. In children and adolescents with ADHD, the MGBA is involved in the pathophysiological mechanisms of neuroinflammation and oxidative stress that give rise not only to the ADHD core symptoms, but also to associated comorbidities, such as sleep disorders. Furthermore, changes in the gut microbiome may also constitute the basis for the efficacy of new alternative therapies currently under investigation, such as omega-3 PUFAs. Thus, the gut microbiome could represent a potential therapeutic target in children and adolescents with ADHD.

Nevertheless, current studies still offer heterogeneous results due to substantial methodological differences in relation to sample size, participant selection criteria, neuropsychological assessment of participants, identification of confounding factors, microbiome analysis techniques, dosage and combination of probiotics, and duration of the intervention period. This could be improved by conducting studies that: include a formal calculation of the required sample size and clearly defined recruitment procedures, including information on the intake of ADHD medication and dietary patterns; use standardized psychometric scales to evaluate ADHD symptoms in accordance with the current diagnostic criteria (DSM-5); have designs that allow inferring causality; use high-resolution techniques beyond 16S rDNA sequencing for taxonomy and functional analysis, such as whole-metagenome shotgun sequencing, metatranscriptomics and metaproteomics. Future research lines should be focused on the design of protocols to standardize the supplementation with probiotics in pediatric populations: composition, dose and time of supplementation.

## Figures and Tables

**Figure 1 nutrients-13-00249-f001:**
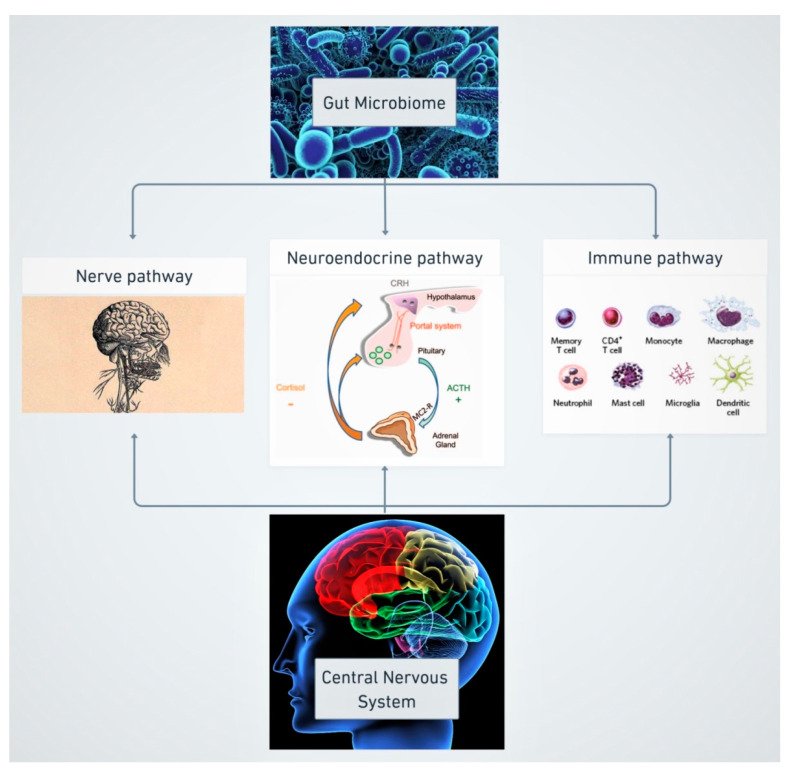
Schematic diagram of the three pathways (nerve, neuroendocrine and immune) that configure the microbiome–gut–brain axis. CRH: corticotropin-releasing hormone; ACTH: adrenocorticotropic hormone; MC2-R: melanocortin receptor 2. CD4+ T cell: CD4-positive T cell.

**Figure 2 nutrients-13-00249-f002:**
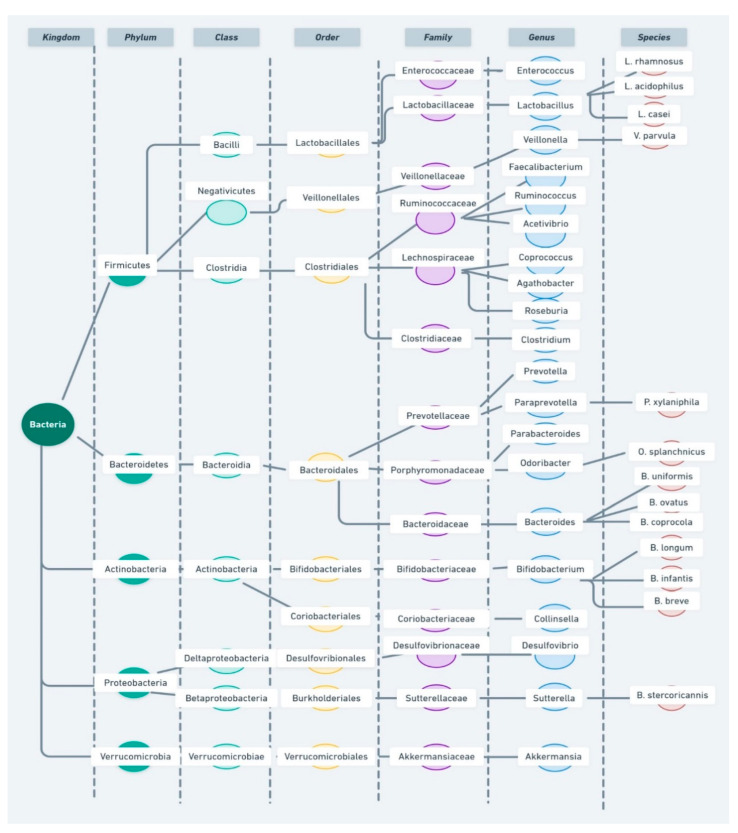
Bacterial taxonomic classification including the genera cited in this article according to the National Center for Biotechnology Information (NCBI) Taxonomy Database [161,162].

**Figure 3 nutrients-13-00249-f003:**
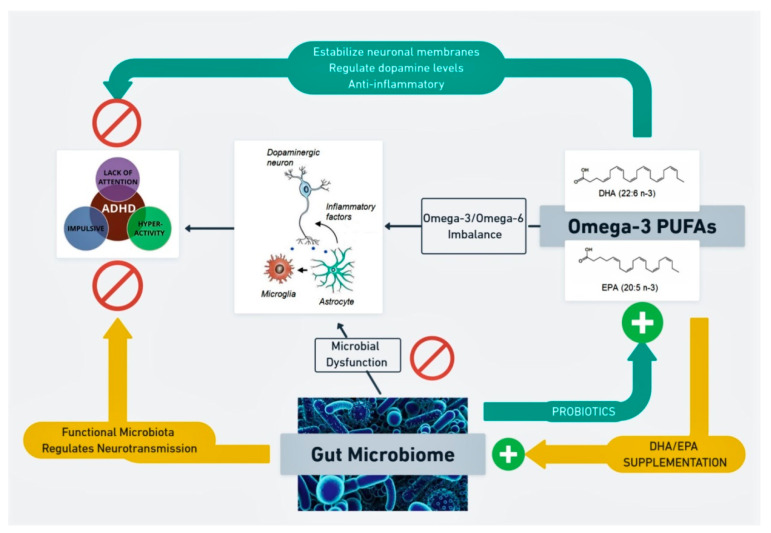
Interaction between the gut microbiome and omega-3 PUFAs (polyunsaturated fatty acids), and effect on ADHD symptoms. The imbalance in the omega-3:omega-6 PUFA ratio leads to a state of neuroinflammation and a consequent disruption in the dopaminergic system (black arrows). Dopaminergic dysfunction is one of the main factors contributting to ADHD core symptoms. Alterations in the microbial composition have also been associated with ADHD symptoms, apparently via the influence of the gut microbiome on neurotransmission (microbiome–gut–brain axis (MGBA)). The administration of probiotics (green arrow) may boost the concentrations of omega-3 PUFAs. This would correct the omega-3:omega-6 imbalance and ultimately improve ADHD symptoms by stabilizing neuronal membranes, dopaminergic neurotransmission and reducing neuroinflammation. On the other hand, supplementation with eicosapentaenoic/docosahexanoic acids (EPA/DHA) (yellow arrow) could induce changes in the microbial composition that would be beneficial for ADHD symptoms, via dopaminergic neurotransmission regulation (MGBA).

**Table 1 nutrients-13-00249-t001:** Characteristics of included studies analyzing differential microbial composition in patients with attention deficit/hyperactivity disorder (ADHD).

Study	Study Characteristics		Potential Confounders	N of Subjects		Results
First Author, Year	Cases	Controls	Potential Confounders	Cases	Controls	
Aarts et al. [133]	Microbiome Sample: ADHD ^1^ cases diagnosed based on DSM-IV^2^ criteria using the K-SADS ^3^.	Healthy participants and unaffected siblings of ADHD participants and self-reported healthy volunteers	Not mentioned	*n* = 19 Age in years, mean (SD ^4^)= 19.5 (2.5) Males = 13 BMI ^5^, mean (SD) = 23.8 (4.1)	*n* = 77 17 healthy participants 21 unaffected siblings of ADHD patients 39 self-reported healthy volunteers Age in years, mean (SD) = 27.1 (14.3) Males = 41 BMI, mean (SD) = 23.0 (3.2)	No significant differences in α ^6^ or β-diversity ^7^. Genus *Bifidobacterium* significantly increased in ADHD patients.
Aarts et al. [133]	fMRI ^8^ sample: from the above ADHD cohort follow-up study: children with ADHD no longer met the diagnostic criteria in adolescence or adulthood.	Healthy and unaffected participants	Not mentioned	*n* = 24 Age in years, mean (SD)= 20.3 (3.7) Males = 18 BMI, mean (SD) = 22.8 (3.5)	*n* = 63 24 healthy participants 39 unaffected siblings Age in years, mean (SD) = 21.3 (3.4) Males = 39 BMI, mean (SD) = 22.7 (2.9)	Decreased ventral striatal response for reward anticipation in patients with ADHD vs. controls (*p* < 0.038)
Aarts et al. [133]	Microbiome and imaging analysis: from the above ADHD cohort	Healthy and unaffected participants		*n* = 6 Age in years, mean (SD)= 18.6 (2.5) Males = 4 BMI, mean (SD) = 22.1 (4.4)	*n* = 22 9 healthy participants 13 unaffected siblings Age in years, mean (SD) = 21.3 (3.3) Males = 13 BMI, mean (SD) = 23.4 (3.7)	Predicted CDT ^9^ relative abundance significantly associated with reward anticipation responses in ventral striatum (standardized beta = −0.42; *p* = 0.048)
Jiang et al. [137]	Juvenile patients diagnosed according with the DSM-IV criteria using the Kiddie-SADS-PL Scale ^10^. ADHD symptom severity assessed via CPRS ^11^.	Healthy control group recruited via advertisement and assessed through a semi-structured clinical interview to exclude individuals with physical illness	Excluded confounders: children with dietary habits; use of probiotics or antibiotics during the 2 months prior sample collection; apparent gastrointestinal symptoms, depressive or anxiety symptoms, obesity, common childhood atopic diseases and/or history of current ADHD medication intake	*n* = 51 treatment-naïve ADHD patients Age in years, mean (SD)= 8.47 (0.47) Males = 38 BMI, mean (SD) = 16.4 (2.02)	*n* = 32 Age in years, mean (SD) = 8.5 (8.47) Males = 22 BMI, mean (SD) = 16.09 (2.02)	No significant differences in α or β-diversity. Significantly lower concentration of the genus *Faecalibacterium* in ADHD patients. Abundance of *Faecalibacterium* was negatively associated with parental reports of ADHD symptoms
Szopinska-Tokov et al. [140]	Adolescents and adults diagnosed in accordance with the DSM-IV criteria via the K-SADS. ADHD symptom severity assessed via CPRS.	Subthreshold ADHD group and healthy control group (composed of siblings of ADHD patients). They were assessed through a semi-structured clinical interview.	Controlled confounders: ADHD medication intake; relatedness factors (unaffected siblings). No information on lifestyle, dietary patterns or antibiotic intake.	*n* = 42 Age in years, mean (SD) = 20.2 (4.2) Males = 26 BMI, mean (IQR) = 23 (20–26)	Control: *n* = 47 Age in years, mean (SD) = 20.5 (3.5) Males = 36 BMI, mean (IQR) = 22 (20–24) Subthreshold ADHD: *n* = 15 Age in years = 20.2 (3.3) Males = 6 BMI, mean (IQR) = 22 (20–25)	No changes in α-diversity, but β-diversity was significantly higher in ADHD patients. A significant increase of the genus *Ruminococcaceae_UGC_004* was detected in the ADHD group (*p* < 0.002), which was associated with inattention scores.
Prehn-Kristensen et al. [141]	All patients met the DSM-IV criteria for ADHD. Measures: German translation of the Kiddie-SADS-PL Scale; CCBCL ^12^; German ADHD rating scale (FBB-HKS ^13^).	*n* = 6. Patients fulfilled criteria for comorbid oppositional defiant disorder (ODD) *n* = 10. Medicine for more than 1 year to treat ADHD symptoms *n* = 9. Medicine for at least 48 h prior to sample.		*n* = 14 Age in years, mean (SD) = 11.9 (2.5) BMI, mean (SD) = 19.0 (3.9) Males = 14	*n* = 17 Age in years, mean (SD) = 13.1 (1.7) BMI, mean (SD) = 18.0 (2.5) Males = 17	α-diversity significantly decreased in ADHD patients vs. controls. β diversity differed significantly between cases and controls. The genus *Neisseria* was identified as marker of the ADHD group. At the family level, a significantly higher abundance of *Bacteroidaceae* was found in cases.
Wan et al. [143]	Diagnosed in accordance with the DSM-5 ^14^ diagnostic criteria via Kiddie-SADS-PL Scale. ADHD symptom severity assessed via CPRS.	Healthy controls recruited via advertisement and assessed through a semi-structured clinical interview.	Excluded confounders: special diets; anxiety or depressive symptoms; digestive diseases; allergic diseases; use of probiotics in the month prior to sample collection; obesity. No data on ADHD medication intake.	*n* = 17 Age in years, median (25th–75th percentiles)= 8 (7,10) BMI, mean (SD) = 16.1 (1.2) Males = 14	*n* = 17 Age in years, median (25th–75th percentiles)= 8 (7, 9.5) BMI, mean (SD) = 15.9 (1.1) Males = 13	No differences in α-diversity. At the genus level, *Faecalibacterium* and *Veillonellaceae* were significantly lower in ADHD patients, whereas *Odoribacter* and *Enterococcus* were significantly higher. The KEGG^15^ revealed significant differences in the DA^16^ and 5-HT ^17^ metabolic pathways.
Casas et al. [150]	ADHD assessed through German parent-completed SDQ ^18^ (10 years) and self-completed version of SDQ (15 years)		Controlled confounders: parental education; indoor factors, e.g., indoor smoking. Reporting bias. No information on medication intake.	*n* = 37 Males = 22 Hyperactivity/inattention: 10 years old = 5 15 years old = 37	*n* = 189 Males = 95 Hyperactivity/inattention: 10 years old= 18 15 years old= 13	Early life bacterial diversity was inversely associated with hyperactivity/inattention at the age of 10 [bacterial OTUs^19^ (medium vs. low: aOR ^20^= 0.4, 95% CI =(0.2–0.8)) and Chao1 (medium vs. low: 0.3 (0.1–0.5); high vs. slow: 0.3 (0.2–0.6)], fungal diversity was directly associated [Chao1 (high vs. low: 2.1 (1.1–4.0)), Shannon (medium vs low: 2.8 (1.3–5.8)), and Simpson (medium vs low: 4.7 (2.4–9.3))]. At the age of 15, only Shannon index was significantly associated with hyperactivity/inattention [bacteria (medium vs. low: 2.3 (1.2–4.2)); fungi (high vs. low: 0.5 (0.3–0.9))].
Stevens et al. [151]	Micronutrient Treatment Group. Cases diagnosed with ADHD via ADHD-RS-IV ^21^.	Placebo Treatment Group Controls diagnosed with ADHD via ADHD-RS-IV.	Not mentioned	*n* = 10 Age in years, mean (SD) = 9.3 (1.3) BMI, mean (SD) = 16.6 (3) Males = 10	*n* = 7 Age in years, mean (SD) = 10.29 (1.9) BMI, mean (SD) = 19.39 (2.9) Males = 7	No changes in α or β-diversity. OTUs significantly increased in the treatment group. Low abundance of Bifidobacterium was associated with low ADHD-RS-IV scores, which is contradictory to the general trend observed in the pre-supplementation and placebo groups.
Wang et al. [153]	Patients with ADHD treated in the outpatient department of a Child Psychiatry. ADHD cases were diagnosed based on DSM-IV-TR ^22^ through structured interview based on the Chinese version of K-SADS-E ^23^. Dietary patterns through food frequency questionnaire.	Children without ADHD. Dietary patterns through food frequency questionnaire.	Excluded confounders: never taken any medications for ADHD; no psychiatric diseases or major physical illnesses. No vegetarians or patients who were currently taking probiotics or antibiotics.	*n* = 30 Age in years, mean (SD) = 8.4 (1.7) Weight in kg, mean (SD)= 30.7 (10.2) Males = 23	*n* = 30 Age in years, mean (SD) = 9.3 (2.2) Weight in kg, mean (SD)= 35.6 (10.6) Males = 18	Gut microbiota communities in ADHD patients showed a significantly higher Shannon Index (*p* = 0.0378) and Chao Index (*p* = 0.0351) than the controls. Simpson Index was significantly lower in ADHD patients.
Cheng et al. [154]	Diagnosed in accordance with DSM ^24^ criteria			*n* = 19,099	*n* = 34,194	*Desulfovibrio* was associated with ADHD.

^1^ Attention-deficit/hyperactivity disorder; ^2^ Diagnostic and Statistical Manual of Mental Disorders 4th Edition; ^3^ Kiddie Schedule for Affective Disorders and Schizophrenia; ^4^ Standard deviation; ^5^ Body mass index; ^6^ Alpha diversity; ^7^ Beta diversity; ^8^ Functional magnetic resonance imaging; ^9^ Cyclohexadienyl dehydratase; ^10^ Kiddie Schedule for Affective Disorders and Schizophrenia, Present and Lifetime Version; ^11^ Conners Parent Rating Scales; ^12^ Child Behavior Checklist; ^13^ Fremdbeurteilungsbogen für hyperkinetische Störung; ^14^ Diagnostic and Statistical Manual of Mental Disorders 5th Edition; ^15^ Kyoto Encyclopedia of Genes and Genomes; ^16^ Dopamine; ^17^ Serotonin; ^18^ Strength and Difficulties Questionnaire; ^19^ Operational taxonomic units; ^20^ Adjusted odds ratio; ^21^ ADHD Rating Scale IV; ^22^ Diagnostic and Statistical Manual of Mental Disorders, 4th Edition, Text Revision; ^23^ Kiddie Schedule for Affective Disorders and Schizophrenia, Epidemiologic Version; ^24^ Diagnostic and Statistical Manual of Mental Disorders.

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
