# Peer review of "Current Evidence on the Role of the Gut Microbiome in ADHD Pathophysiology and Therapeutic Implications"

_nutrients, 2021, doi:10.3390/nu13010249_

Round 1

Reviewer 1 Report

The review by Checa-Ros et al., is well-written and covers a topic of great interest of this audience. In order to improve the text this reviewer have the following recommendations:

The references discussed with regards to the gut-brain axis field are a little out-dated and its current state this review does not add to the many reviews in the field . Much has been developed on the link of gut microbiota,neurodevelopment and mood disorders. This review should include some of the most recent findings in the field.

The session on Gut Microbiome and Neurodevelopment: the Gut-Brain-Axis should be expanded.

Author Response

Reviewer 1

We thank this reviewer for his/her overall positive assessment of our work and his/her recommendations to produce a more updated and comprehensive review manuscript.

Point 1: The references discussed with regards to the gut-brain axis field are a little out-dated and its current state this review does not add to the many reviews in the field. Much has been developed on the link of gut microbiota, neurodevelopment and mood disorders. This review should include some of the most recent findings in the field.

Response 1: We have extensively revised section 2 (Gut Microbiome and Neurodevelopment: the Gut-Brain Axis) and updated a substantial number of the references between 28 and 103, mainly including studies published between 2017 and 2020. In order to describe the link between gut-brain-axis, neurodevelopment and mood disorders in a more clear way, we have added recent investigations in this field: Schwarz et al. 2018 (ref 48); Mathee K. et al, 2020 (ref 49); Chang et al. 2020 (ref 52); Zhu et al. 2020 (ref 54); Hwang et al. 2019 (ref 61); Li et al. 2020 (ref 63); Xiao et al. 2020 (ref 88). 

Point 2: The session on Gut Microbiome and Neurodevelopment: the Gut-Brain-Axis should be expanded.

Response 2: We have described each paragraph of this section in more detail and included references to the role of the gut microbiome in: the enteric nervous system (lines 93 to 103); the kynurenine pathway (lines 124 to 135); gastrointestinal peptides (lines 216 to 223); and the opioid system (lines 229-231).

We reiterate our thanks and hope to have addressed all your comments.

Reviewer 2 Report

Comments: Authors are addressing an important area (MBGA) of research. Adding ADHD in MBGA makes it more interesting.

  1. Figure1 has a lot of empty space. It would be great to increase those small figures to a little bit bigger. Immune cells are also not clear. 
  2. The visual impact is very important, especially in the review articles. I suggest making a figure to show the relationship between the gut microbiome, ADHD, ω-3 PUFAs, and addressing MGBA. This will help a lot to the readers.
  3. Line 205-206 , "Pathophysiological Implications" is not in italics and missed out from 3.2 section.

Author Response

Reviewer 2

We really appreciate the suggestions made by this reviewer, which has contributed to improve the visual impact of our manuscript.

Point 1:

Figure 1 has a lot of empty space. It would be great to increase those small figures to a little bit bigger. Immune cells are also not clear.

Response 1:

The images have been changed for clearer ones, and the figure has been oriented vertically instead of horizontally.

Point 2:

The visual impact is very important, especially in the review articles. I suggest making a figure to show the relationship between the gut microbiome, ADHD, ω-3 PUFAs, and addressing MGBA. This will help a lot to the readers.

Response 2:

A new figure (Figure 3) at the end of section 3.3 has been added with this objective.

Point 3:

Line 205-206, "Pathophysiological Implications" is not in italics and missed out from 3.2 section.

Response 3:

Change made.

We hope to have successfully addressed all your comments.

Reviewer 3 Report

This topic is currently hot and there is a number of recent review articles published on it. Some reviews have been conducted in a more systematic way and can provide more solid conclusions, pointing out the weaknesses of existing studies.

See for instance

Prog Neuropsychopharmacol Biol Psychiatry. 2020 Dec 1;110187. doi: 10.1016/j.pnpbp.2020.110187.

Nutrients. 2020 May 28;12(6):1573. 

Front Psychiatry. 2020 Jun 26;11:623.

In any case, reviews on the present topic can draw evidence only from a very limited number of adequately conducted experimental studies. The evidence base for these review articles is still inconsistent and reviews, especially when narrative as in the present case, cannot help clarifying the topic.

The present review reports on interesting hypotheses and potentially promising studies. However, real conclusions cannot be drawn due to the scarcity of data and the contradictory nature of study results.

Thus, the present review would be much improved by implementing more methodology and criticism when considering each study presented. Suggestions on how to improve the methodology of this review and its contribution to the field could be:

A) highlight, every time for each piece of literature presented, what are the limitations, in brief, proving a judgment on quality and reliability

B) add tables or structured paragraphs, highlighting what is consistent and what is controversial across studies. This may be done separately regarding pathophysiological associations with ADHD (healthy vs. ADHD) and regarding therapeutic strategies (treated ADHD vs. non treated ADHD).

C) a new paragraph could be required, reporting an overall criticism of current studies, highlighting what the methodological limitations are, and giving suggestions on how to improve future research in the field.

Given the current evidence base, it is methodologically impossible to draw any conclusion on the objectives 2,3,4,5 proposed for this review. It is therefore important to analyze systematically why this is not possible and to suggest solutions.

In the present version of this manuscript, only a narrative review and criticism is given. I am afraid that more and more non-systematic reviews risk adding confusion to the field.

Author Response

Reviewer 3

We thank this reviewer for his/her insightful comments and accept the weaknesses of our work. Our review was essentially narrative, and although we tried to do a careful revision of the most recent literature on the topic, we admit we were not able to add new evidence or provide much clarification to the current state of knowledge. Certainly, a systematic review or meta-analysis would have increased the impact of our paper, and this is something we will take on board for future publications. Therefore, in this context, we welcome the suggestions made by the reviewer as a way to enhance the quality and consistency of our manuscript.

Point 1:

A)highlight, every time for each piece of literature presented, what are the limitations, in brief, proving a judgment on quality and reliability

Response 1:

In points 3.2 (microbiome and ADHD pathophysiology), 3.3 (microbiome and omega-3 PUFAs), 3.4 (microbiome and sleep disorders) and 3.5 (points): for each new piece of literature, we have added (track changes) a more detailed description of the limitations and a brief judgment on the overall quality and reliance (recruitment of participants, representativeness/comparability of samples, design, novelty of the study).

Point 2:

  1. B) add tables or structured paragraphs, highlighting what is consistent and what is controversial across studies. This may be done separately regarding pathophysiological associations with ADHD (healthy vs. ADHD) and regarding therapeutic strategies (treated ADHD vs. non treated ADHD).

Response 2:

We have included a structured paragraph in sections 3.2 (lines 611-645) and 3.5 (lines 977-999), trying to show consistencies across studies and contrasting results.

Point 3:

  1. C) a new paragraph could be required, reporting an overall criticism of current studies, highlighting what the methodological limitations are, and giving suggestions on how to improve future research in the field.

Response 3:

This paragraph has been included in the last section (5. Concluding Remarks, lines 1042-1055), highlighting considerable divergences between studies, the main limitations in methodology and making suggestions to address this issue.

Point 4:

Given the current evidence base, it is methodologically impossible to draw any conclusion on the objectives 2,3,4,5 proposed for this review. It is therefore important to analyze systematically why this is not possible and to suggest solutions.

Response 4:

We have added a new section (section 4. Limitations) to emphasize the limitations of our study and how this could be improved with a different design.

We hope to have successfully addressed all your comments.

Round 2

Reviewer 3 Report

Revisions made have improved significantly the manuscript.